# An appraisal of fund of funds efficiency based on risk-adjusted performance measures: Application of an augmented WASPAS methodology

Mostafa Shabani[1], Ali Khodarahmi[2], Rouzbeh Ghousi[1]*, Emran Mohammadi[1], Hossein Ghanbari[1]

**1** Department of Industrial Engineering, Iran University of Science and Technology, Tehran, Iran,
**2** Department of Management, University of Tehran, Tehran, Iran

\* ghousi@iust.ac.ir

## Abstract

In the complex landscape of investment management, Fund of Funds play a crucial role in constructing diversified portfolios with professional oversight. However, evaluating the performance of Fund of Funds is a challenging task due to their intricate management layers and diverse underlying assets. Traditional performance assessment methodologies often fall short in capturing these complexities, leading to potential inefficiencies in decision-making. This study addresses these challenges by applying a Multi-Attribute Decision Making approach, specifically an augmented Weighted Aggregated Sum Product Assessment methodology, to provide a comprehensive appraisal of Fund of Fund performance. This study introduces an innovative modification to the Weighted Aggregated Sum Product Assessment methodology, addressing its inherent limitations and providing a more advanced evaluative framework. Despite its ability to offer a structured basis for decision-making, this approach has certain constraints, particularly in identifying the most influential performance criteria impacting outcomes. To resolve this issue, the proposed modification incorporates entropy measures and effective weighting techniques, thereby enhancing the robustness and precision of performance evaluation. This approach provides a sophisticated evaluative framework, facilitating investors in making informed decisions and optimizing investment strategies. This methodology was tested on a set of Iranian Fund of Funds, with computational results demonstrating the strong performance and robustness of the proposed approach. Among the alternatives evaluated, TMSK emerged as the best-performing fund, highlighting the effectiveness of the modified Weighted Aggregated Sum Product Assessment method in facilitating informed investment decisions and optimizing portfolio strategies.

**Data availability statement:** All investment fund data files are available from https://fipiran.com/ and Supporting information.

**Funding:** The author(s) received no specific funding for this work.

**Competing interests:** The authors have declared that no competing interests exist.

## 1. Introduction

Investment can drive economic growth and development by generating returns on resources used during production. However, achieving long-term and continuous economic growth requires the optimal allocation and allocation of resources at the level of the national economy, and this is not possible without the help of financial markets, especially the extensive and efficient capital market [1]. Broadly, investment methods can be categorized into direct and indirect investments in the stock exchange. Investing in mutual funds is a well-known indirect investment strategy. Mutual funds attract investors by offering benefits such as diversification, reduced unsystematic risk, professional management, liquidity, and economies of scale. Fund of Funds (FOFs) are a specialized type of alternative investment fund that allocates capital to other fund managers. This is as opposed to investing directly in securities of public companies as many hedge funds do, fund of funds are investment vehicles that allocate capital to other fund managers. The way a fund of funds works in practice is that investors allocate capital to the fund of funds manager. The fund of funds manager then divides up the pool of capital from investors and allocates it among different fund managers. The fund managers that receive this capital are commonly called underlying managers or sub-advisers. There are two common types of fund of funds. The first is a hedge fund of funds, sometimes called a fund of hedge funds, that allocates capital underlying hedge fund managers. The second type is a private equity fund of funds. As the name implies, this type of fund allocates capital to underlying private equity managers [2]. Among its other types, we can mention mutual fund FOF, investment trust FOF, and VC FOF. A fund-of-funds entity can be organized in several ways, including as a limited partnership (LP), a limited liability company (LLC), an offshore corporation, or a trust. Depending on the structure, the type of interest held by an investor will differ. Fund of Funds offer several distinct advantages that make them an attractive option for investors. Primarily, these benefits revolve around diversification, due diligence, risk management, portfolio management, access to funds, consolidated reporting, and performance enhancement [3].

Fund of Funds achieve diversification by allocating their assets across multiple hedge funds, encompassing a variety of investment strategies and sub-strategy approaches. This inherent diversification reduces the risk of investing in any single hedge fund. Professional due diligence is another critical benefit. Funds of funds meticulously assess hedge fund managers through a comprehensive process that gathers all available information, verifies its accuracy, and evaluates the results. This thorough scrutiny ensures that only the most reliable and promising managers are selected. Risk management is an area where funds of funds particularly excel. They provide rigorous risk control for each fund within their portfolio and the overall portfolio itself. This expert oversight in risk management is highly valued by investors, who gain confidence knowing that their investments are being carefully monitored. Regarding portfolio management, funds of funds aim to generate returns that exceed industry averages by strategically selecting or emphasizing certain hedge fund strategies expected to outperform, while avoiding or minimizing exposure to those

anticipated to underperform. This selective approach seeks to maximize returns and minimize risks. Access to otherwise inaccessible hedge funds is another notable advantage. Funds of funds enable investors to invest in hedge funds that may be out of reach due to capital limitations or those closed to new investments. This expanded access broadens the investment opportunities available to investors. Additionally, funds of funds streamline the administrative burden by consolidating performance data from all underlying funds each month. This consolidated reporting simplifies the monitoring process for investors. Ultimately, the performance of funds of funds often aims to outperform the broader hedge fund industry. A key benefit is their ability to generate alpha, offering returns above those of a passive index allocation, thereby enhancing the overall performance of an investor's portfolio.

When considering the disadvantages of investing in a fund of funds, several critical factors must be weighed against their benefits. One major disadvantage involves exposure to other investors' cash flows. Typically, a fund of funds pools the assets of various investors and collectively invests this money in hedge funds. However, the handling of investor inflows and outflows by the fund of funds may not always be optimal, potentially disadvantaging some investors. Another significant drawback is the additional layer of fees and expenses imposed by funds of funds, which are charged on top of those levied by the underlying hedge funds. This extra cost is often viewed negatively by investors. The lack of control and customization is also a notable disadvantage. Investors have limited influence over the investment decisions within a fund of funds. To mitigate this, some investors choose to diversify by investing in multiple funds of funds, seeking a broader spread of investments. Lastly, decreased transparency is a concern. Investing through a fund of funds means that the investor does not have a direct relationship with the individual hedge funds. The transparency provided by each hedge fund is relayed through the fund of funds, leading to a more opaque investment process compared to direct hedge fund investments.

As seen in Fig 1, Mutual fund types can be classified based on various characteristics such as asset class, investment goals, structure, risk, etc. Currently, the types of investment funds in Iran are equity funds, debt funds, exchange-traded commodities (ETC), blend funds, venture capital, private equity, business financial management (BFM), construction funds, real estate investment trusts (REITs), and FOFs. Funds that can be traded in the stock market are called ETFs, otherwise, they are called creation and redemption.

In the intricate and evolving investment milieu, Fund-of-Funds are indispensable for constructing diversified, professionally managed portfolios. However, the evaluation of FOF performance presents significant challenges due to the multi-layered management structures and heterogeneous underlying assets. Conventional assessment methodologies often fail to encapsulate these complexities. This article seeks to bridge this gap by employing Multi-Attribute Decision Making (MADM) techniques to conduct a comprehensive appraisal of FOF performance, incorporating both financial and non-financial dimensions. The assessment of FOFs is inherently complex due to their stratified structure and diverse asset composition. Traditional performance metrics frequently prove inadequate, resulting in suboptimal decision-making. There exists a critical need for an evaluative framework that amalgamates both qualitative and quantitative criteria for a holistic assessment.

This study endorses the implementation of a multicriteria approach, specifically the Weighted Aggregated Sum Product Assessment (WASPAS) methodology, to establish a robust framework for FOF evaluation. WASPAS amalgamates the weighted sum model (WSM) and the weighted product model (WPM), thereby furnishing a meticulous and reliable evaluation by synergizing the straightforwardness of additive methods with the precision of multiplicative approaches. However, despite its ability to provide a structured decision-making framework, the WASPAS method has certain limitations. Specifically, it struggles to determine the best performance criteria affecting outcomes. To overcome this limitation, this study introduces a novel modification to the WASPAS method, which uses entropy and effective weighting. The augmented WASPAS approach significantly refines traditional WASPAS by incorporating entropy-based weighting, which provides a more objective and precise weighting of criteria, thus reducing bias and amplifying the impact of the most informative criteria on the decision-making process. This enhanced sensitivity to performance differentials across multiple attributes

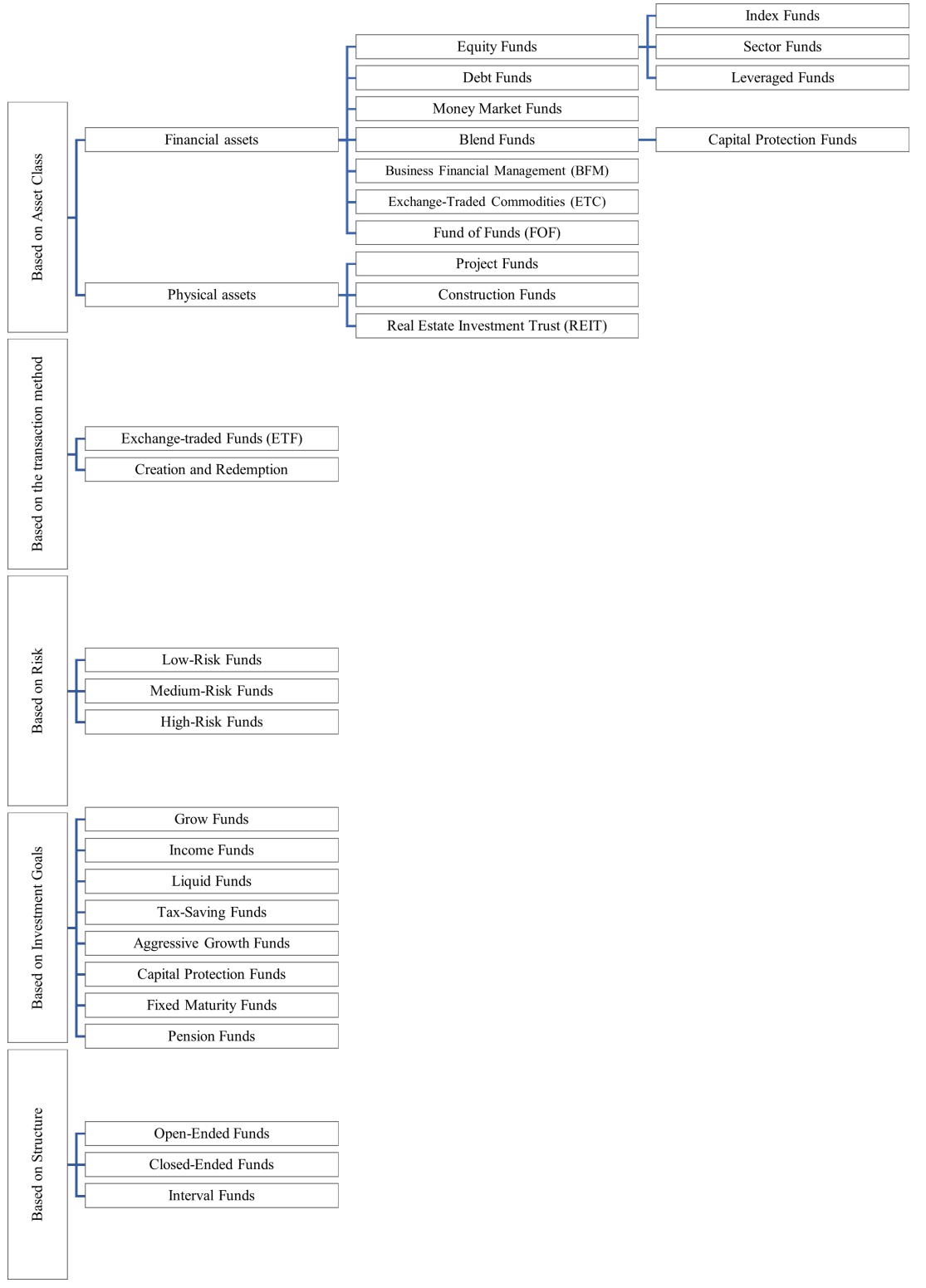

**Fig 1. Types of investment funds with different bases.**

overcomes the limitations of the original method, enabling a more sophisticated evaluation of complex investment portfolios. This enhanced approach provides a more sophisticated evaluative framework, enabling investors to make informed decisions and optimize their investment strategies.

The remainder of this paper is structured as follows: Section 2 provides a comprehensive overview of recent scholarly work on FOF evaluation and the WASPAS methodology, grounded in an exhaustive literature review. Section 3 elucidates the proposed application of the WASPAS method. Section 4 delineates the alternatives and specifies the factors and criteria essential for evaluation. Section 5 integrates the results and discusses the implications of implementing the WASPAS methodology. Finally, Section 6 presents the conclusions and proffers recommendations for future research directions.

## 2. Literature survey

This section provides a concise yet comprehensive review of recent research on evaluating the sustainability of Fund of Funds (FoF) investments, with a particular focus on the application of the WASPAS methodology. The literature survey is organized into two subsections. Section 2.1 is devoted to a critical review of the literature on Fund Analysis and Fund of Funds Investments, examining key studies and methodologies in this domain. Section 2.2 then offers an in-depth exploration of the WASPAS methodology, discussing its application and relevance in the context of FoF performance evaluation.

### 2.1. Literature review of fund analysis and fund of funds investment strategies

Guo and Shen [4] analyzed the performance and sustainability of China's retirement Fund of Funds. Their study utilized standard relative and absolute measures such as the Sharpe ratio, Treynor ratio, and Jensen's alpha to assess performance. They also evaluated the sustainability of performance using performance dichotomy, cross-sectional regression, and Spearman rank correlation coefficient methods. The analysis revealed that FOFs, particularly the aggressive and stable groups, generated higher returns and exhibited sustainability, especially during economic recessions. The stable group of funds notably produced greater investment returns, with all statistically significant alpha values for Jensen being positive. This study highlights the benefits of diversification in FOFs.

Zhang [5] compares various risk models and concludes that Funds of Funds offer effective risk diversification and stable returns, particularly through the risk parity model. By using selected performance evaluation indicators, the study visually compares the risk parity model, equal volatility model, and equal weight model in terms of risk and return. The findings demonstrate that the risk parity model excels in providing good risk diversification and strong returns. Wolf and Wunderli [6] used advanced statistical techniques to select hedge funds for FOFs, finding that portfolios constructed using these methods have attractive return properties compared to simple equal-weighting strategies. Bertin and Prather [7] report on FOFs' characteristics and performance relative to traditional equity mutual funds and find that FOFs compare favorably. FOFs with identified managers outperform their unidentified counterparts, and FOFs that invest in-family outperform both traditional equity funds and those FOFs investing out-of-family. Finally, replicating FOFs' holdings can be prohibitively expensive since they commonly hold funds with high minimum initial investments, closed funds and/or funds that are restricted to a particular investor type.

Brands and Gallagher [8] examined the performance and diversification properties of active Australian equity FOFs, demonstrating that increased diversification within FOFs enhances portfolio performance and reduces risk, particularly in mean-variance settings. Using simulation analysis, the study finds that as the number of funds in a FOF portfolio increases, performance improves, though skewness and kurtosis measures may be less favorable depending on investor preferences. The majority of diversification benefits are realized when a portfolio includes approximately six active equity funds. Antypas et al. [9] evaluate the Morningstar mutual fund ranking system. We find that indeed higher Morningstar ratings are associated with higher returns on the portfolios including respectively five-, four-, three, two- and one-star funds only (STAR5 to STAR1). We then perform an unconditional and conditional portfolio performance evaluation. In both cases the evidence suggests that the better performance of the STAR3, STAR4 and STAR5 categories reflects superior

stock selection rather than market timing abilities. Overall, the implication for the Morningstar ranking system is that this is most effective in identifying the worst-performing funds (STAR1 or STAR2) rather than the best-performing ones. Evans and Sun [10] examined the role of factor models and simple performance heuristics in investor decision-making using Morningstar's 2002 rating methodology change. Their results imply that improvements in simple performance heuristics can result in more sophisticated risk adjustment by retail investors.

Tabrizi and Gharehjeloo [11] scrutinized the performance of mutual funds within the Iranian Capital Market by integrating market timing models with the Fama and French three-factor model. The findings indicate an absence of statistically significant market timing prowess and security selection among all evaluated cases. Nevertheless, a positive, statistically significant influence of size and book-to-market ratio was detected in one and three mutual funds, respectively. The panel model results revealed a negative, statistically significant size effect and security selection, alongside a positive, statistically significant beta and book-to-market ratio, with no discernible market timing ability in either the Treynor-Mazuy or Henriksson-Merton models. Contrary to traditional models, the combined models yielded superior outcomes.

Agarwal and Mirza [12] addressed multiple issues including measuring the performance of selected mutual schemes based on risk and return. The study compares the performance of these selected schemes with benchmark indices to determine if they are outperforming or underperforming. Additionally, it ranks funds based on performance and suggests investment strategies.

Sadeghi Moghadam et al. [13] scrutinized the assessment of selected mutual funds within the Iranian stock market through an amalgamation of TOPSIS, VIKOR, and a similarity-based approach. This appraisal endeavors to juxtapose three categories of indices, encompassing general evaluation indices (age, net asset value of the mutual funds, percentage of cash assets, and net asset value), risk-adjusted evaluation indices (Sharpe, Treynor, and Jensen), and risk-adjusted evaluation indices utilizing semivariance (modified Sharpe, modified Treynor, and modified Jensen with downside Beta), both independently and collectively, to gauge the efficacy of these methodologies with the varied indices. Ultimately, the outcome of this comparative analysis is presented, evaluating the decisions made by novice and seasoned investors.

Yuan and Yuan [14] propose a comprehensive multi-index ranking method for mutual fund performance, addressing the limitations of existing ranking approaches. Traditional single-index methods are often inadequate, while multi-index methods tend to be indirect and inflexible. The new method combines both cardinal and ordinal data, offering a more accurate and robust ranking. It also introduces a novel objective weighting approach to address index correlation issues. This flexible and practical method provides a clearer and more reliable evaluation of mutual fund performance, making it a valuable tool for investors and researchers.

Malhotra et al. [15] offer a detailed analysis of the performance of financial mutual funds, focusing on a 23-year period significantly longer than previous studies. The research incorporates multiple performance evaluation techniques, such as risk-adjusted returns, multifactor analysis, and market timing, providing a holistic view of fund behavior. The findings indicate that financial mutual funds outperform both the market and sector-specific benchmarks in terms of risk-adjusted returns, though their alphas remain indistinguishable from zero. Moreover, the study reveals that fund managers do not possess market timing or security selection skills. The authors further dissect performance trends across different periods, including the pre- and post-2008 financial crisis and the recent COVID-19 pandemic, offering insights into the stability and adaptability of financial mutual funds under various market conditions.Top of FormBottom of Form

Mota et al. [16] introduce a new hybrid multiple-criteria decision-making (MCDM) methodology, BWM-Moora-N, for prioritizing investment funds. The study combines the Best-Worst Method (BWM) with the Multi-Objective Optimization by Ratio Analysis (MOORA) to address the complexities inherent in evaluating and selecting investment funds. This method overcomes the limitations of traditional MCDM approaches by improving accuracy in the prioritization process, particularly in scenarios involving a large number of criteria. The authors demonstrate the effectiveness of the proposed model through empirical analysis, showing how it provides a more structured and reliable decision-making framework for investors, thereby enhancing the strategic selection of investment funds in diverse market conditions.

## 2.2. The application of WASPAS methodology in research studies

The WASPAS paradigm epitomizes a significant shift in Multi-Attribute Decision Making, integrating the Weighted Sum and Product Models to overcome traditional limitations. This approach enhances decision-making precision and robustness, enabling the effective evaluation of diverse criteria and achieving widespread adoption across various domains.

Fox [17] characterizes MADM as a structured method for evaluating and ranking options across multiple criteria. Widely applied across various fields, this approach strengthens decision-making by thoroughly considering each pertinent attribute of the available choices. Among MADM techniques, the WASPAS framework is notable for its dual-weighted mechanism, which enables a more refined aggregation of criteria, facilitating precise prioritization and selection among alternatives.

Konstants [18] emphasizes that utilizing the MADM framework, especially through the advanced capabilities of the WASPAS method, equips investors and decision-makers with a structured approach for systematically assessing and comparing investment options. By synthesizing multiple financial indicators and risk parameters, this rigorous analysis facilitates the precise identification of optimal investment prospects, maximizes potential returns, and strengthens risk management strategies.

Zavadskas et al. [19], the architects of the WASPAS methodology, have introduced a distinguished framework in Multi-Attribute Decision Making. This methodology is renowned for its unique capability to amalgamate both qualitative and quantitative criteria, thereby enhancing decision-making processes through a more comprehensive analysis. Traditional MADM methodologies, which often rely on intricate utility functions, pose significant challenges for some decision-makers. In contrast, the WASPAS methodology seamlessly integrates the WSM and the WPM, making it more accessible and user-friendly. The WASPAS methodology fundamentally roots itself in the confluence of the WSM and the WPM. The WSM aggregates the weighted performance scores of each alternative across all criteria, while the WPM multiplies the performance scores raised to the power of their respective weights. By synergizing these two models, WASPAS leverages the strengths of both additive and multiplicative aggregation techniques, ensuring a robust and comprehensive evaluative process. This integration not only simplifies the decision-making process but also enhances its accuracy and reliability.

Aliyev [20] elucidates the genesis of WASPAS, an innovative methodology conceived to transcend the constraints of conventional MADM approaches by furnishing a more robust and pliable framework. This method aspires to amplify decision-making precision and adeptly manage an eclectic array of evaluative criteria. Its steadfast reliability and facile application have precipitated its assimilation into myriad domains over time. WASPAS proffers numerous advantages vis-à-vis other MADM methods, including augmented accuracy through the amalgamation of WSM and WPM, versatility in accommodating a broad spectrum of criteria and data types, and user accessibility, rendering it propitious for diverse applications.

Fox [21] notes that fund evaluation within MADM involves assessing multiple financial metrics such as return on investment, risk levels, management quality, and market conditions. By applying MADM methods like WASPAS, investors can derive a holistic view of each fund's performance, enabling better-informed investment decisions.

Zavadskas and Turskis [22] reports that performance metrics from comparative studies reveal that WASPAS consistently delivers high-ranking accuracy, low computational complexity, and excellent decision stability. These outcomes make it a preferred choice for complex decision-making scenarios like fund evaluation.

Chakraborty and Zavadskas [23] asserts that the efficacy and efficiency of WASPAS in fund evaluation are well-documented. It effectively combines multiple decision criteria into a cohesive evaluation framework, ensuring comprehensive and reliable investment analysis.

Li et al. [24] posit that prospective advancements in the WASPAS methodology should emphasize the incorporation of sophisticated data analytics techniques, the augmentation of computational efficiency, and the assimilation of real-time data to facilitate more dynamic decision-making processes. Furthermore, innovations could investigate hybrid paradigms

that amalgamate WASPAS with other nascent decision-making frameworks. Their comparative analyses reveal that WASPAS frequently surpasses other MADM methodologies such as AHP, TOPSIS, and SAW in terms of decision accuracy and robustness, underscoring its superior capability to synthesize multiple criteria efficaciously.

## 3. The proposed WASPAS method: A thorough step-by-step explication

The WASPAS method synthesizes the Weighted Sum Model and the Weighted Product Model to furnish a robust framework for multi-criteria decision-making. However, despite its ability to offer a structured decision-making framework, the WASPAS method has certain limitations. Specifically, it struggles to determine the best performance criteria affecting outcomes. To address this limitation, this study introduces a novel modification to the WASPAS method, which uses entropy and effective weighting.The augmented WASPAS approach significantly refines traditional WASPAS by incorporating entropy-based weighting, which provides a more objective and precise weighting of criteria, thus reducing bias and amplifying the impact of the most informative criteria on the decision-making process. This enhanced sensitivity to performance differentials across multiple attributes overcomes the limitations of the original method, enabling a more sophisticated evaluation of complex investment portfolios. The integration of the WSM and WPM within the augmented approach strengthens robustness, maintaining ranking stability and reliability even with minor fluctuations in input data. This dual aggregation, enhanced by entropy-driven weights, creates a comprehensive, multidimensional framework that ensures rigorous and dependable assessments of Fund of Funds performance, positioning it as an advanced tool for high-precision, data-driven decision-making in portfolio management. Below is a detailed, scientifically rigorous elucidation of the proposed WASPAS method:

**Step 1:** *Formulation of the Decision Matrix*

Construct the decision matrix $X = [x_{ij}]_{m \times n}$, where:

- $x_{ij}$ denotes the performance metric of the $i_{th}$ alternative with respect to the $j_{th}$ criterion.

- $m$ represents the total number of alternatives.

- $n$ signifies the number of criteria under consideration.

The decision matrix serves as the foundational structure in which all performance metrics are systematically organized. Each element $x_{ij}$ encapsulates the performance of alternative $i$ under criterion $j$, thereby facilitating a comprehensive evaluation of all alternatives across multiple criteria.

**Step 2:** *Normalization of the Decision Matrix*

Normalize the decision matrix elements to render the data dimensionless, ensuring uniform comparability. The normalization technique applied is contingent upon the criterion type (beneficial or non-beneficial).

- For beneficial criteria (where higher values are preferable):

$$\bar{x}_{ij} = \frac{x_{ij} - Min(x_{ij})}{Max(x_{ij}) - Min(x_{ij})}$$

(1)

- For non-beneficial criteria (where lower values are preferable):

$$\bar{x}_{ij} = \frac{Max(x_{ij}) - x_{ij}}{Max(x_{ij}) - Min(x_{ij})}$$

(2)

Normalization is crucial as it standardizes the criteria measurements, eliminating any scale discrepancies and ensuring that each criterion contributes equitably to the decision-making process.

 

**Step 3**: *Entropy-Based Weighting*

Calculate the entropy for each criterion and determine the weights based on the entropy values.

- Calculate the probability matrix $P = [p_{ij}]_{m \times n}$:

$$p_{ij} = \frac{\overline{x}_{ij}}{\sum_{i=1}^{m} \overline{x}_{ij}} \tag{3}$$

- Calculate the entropy $H_j$ for each criterion $j$:

$$H_j = -k \sum_{i=1}^{m} p_{ij} \ln \left( p_{ij} \right) \tag{4}$$

where $k = \frac{1}{\ln(m)}$

- Determine the degree of diversification $d_j$ for each criterion $j$:

$$d_j = 1 - H_j \tag{5}$$

- Calculate the weights $w_j$ for each criterion $j$:

$$w_j = \frac{d_j}{\sum_{j=1}^{n} d_j} \tag{6}$$

**Step 4:** *Computation of the Aggregate Weight Using the Weighted Sum Model (WSM).*

Determine the aggregate weight for each alternative using the Weighted Sum Model:

$$Q_i^{(1)} = \sum_{j=1}^{n} \overline{x}_{ij} w_j \tag{7}$$

where:

$w_j$ denotes the weight assigned to the $j_{th}$ criterion, reflecting its relative importance.

The Weighted Sum Model aggregates the normalized scores, weighted by their respective importance, to produce a composite score for each alternative. This aggregation provides an initial quantitative measure of each alternative's overall performance.

**Step 5:** *Evaluation of the relative importance using the Weighted Product Model (WPM).*

Evaluate the relative importance of each alternative using the Weighted Product Model:

$$Q_i^{(2)} = \prod_{j=1}^{n} \left( \overline{x}_{ij} \right)^{w_j} \tag{8}$$

The Weighted Product Model, on the other hand, multiplies the normalized scores, raised to the power of their respective weights. This multiplicative aggregation highlights the proportional contribution of each criterion, providing a nuanced measure of each alternative's relative importance.

**Step 6:** *Integration of WSM and WPM Scores*

Integrate the scores obtained from WSM and WPM to derive a comprehensive metric for each alternative:

$$Q_i = 0.5 \, Q_i^{(1)} + 0.5 \, Q_i^{(2)} \tag{9}$$

This formula ensures an equitable contribution of both models to the final assessment. By integrating the scores from WSM and WPM, WASPAS synthesizes both additive and multiplicative approaches, offering a balanced and holistic evaluation of the alternatives.

**Step 7:** *Generalized Aggregated Criterion*

Develop a generalized aggregated criterion to allow for an adjustable balance between WSM and WPM:

$$Q_i = \lambda \ Q_i^{(1)} + (1-\lambda)Q_i^{(2)} \tag{10}$$

where $\lambda$ is a weighting parameter that oscillates between 0 and 1, modulating the equilibrium between WSM and WPM. This generalized criterion confers adaptability, empowering decision-makers to fine-tune the balance in accordance with specific decision-making paradigms or predilections. The precision of WASPAS is posited to be evaluated based on the initial criteria accuracy for values of $\lambda$ spanning this interval. When $\lambda$ equals 0, WASPAS metamorphoses into WPM, and when $\lambda$ equals 1, it transmutes into WSM.

**Step 8:** *Selection of the Optimal Alternative*

Identify the alternative with the maximal combined score $Q_i$, which signifies the optimal decision considering all criteria and their respective weights. The final step involves selecting the alternative with the highest combined score $Q_i$, which represents the most favorable decision based on a comprehensive assessment of all criteria and their respective importances. This approach guarantees that the selected alternative is optimal within the defined multi-criteria framework.

## 4. Identification of alternatives and definition of factors and criteria

Investment is a two-dimensional process, including risk and return. One of the basic problems in performance evaluation is the human tendency to focus on the portfolio return and not pay enough attention to the risk incurred to obtain the desired return. The performance should be evaluated relatively and therefore we need an index for comparison. Without considering the risk, it is not possible to examine different investment solutions only through returns. Although all investors prefer higher returns, it should be kept in mind that investors are also risk averse. To properly evaluate the performance of the portfolio, it should be determined whether the return considering the risk is large enough or not. To accurately evaluate the performance, the performance of the portfolio should be evaluated based on adjusted risk. For this purpose, two types of risk can be estimated; The market (systematic) risk of the portfolio, which is measured by beta ($\beta$), and the total risk of the portfolio, which is determined by the standard deviation ($\sigma$). For example, if the evaluated portfolio represents the total risk investment funds, the Sharpe Ratio and $M^2$ measures are appropriate, and if the evaluated portfolio is a part of the total risk investment funds, Jensen's Alpha and Trainer Ratio are more appropriate.

As illustrated in Fig 2, risk-adjusted performance evaluation measures are divided into two primary categories: those grounded in Modern Portfolio Theory and those based on Post-Modern Portfolio Theory [25]. Modern Portfolio Theory focuses on optimizing portfolios by balancing returns with overall risk, typically measured by standard deviation, under the assumption of normally distributed returns. In contrast, Post-Modern Portfolio Theory builds on Modern Portfolio Theory by addressing limitations inherent in the traditional approach, such as its treatment of risk and return symmetry [26]:

- Using adverse risk instead of standard deviation as a risk measurement tool.

- post-modern portfolio theory also includes non-normal return distributions.

The objective of this section is to meticulously delineate the diverse criteria and alternatives pertinent to fund evaluation indicators. Subsequently, the methodologies employed are rigorously examined, assessed, and juxtaposed. Initially, information was meticulously aggregated through an exhaustive literature review and consultations with domain experts specializing in fund evaluation. Leveraging the insights gleaned from the literature and expert opinions, we elucidate the

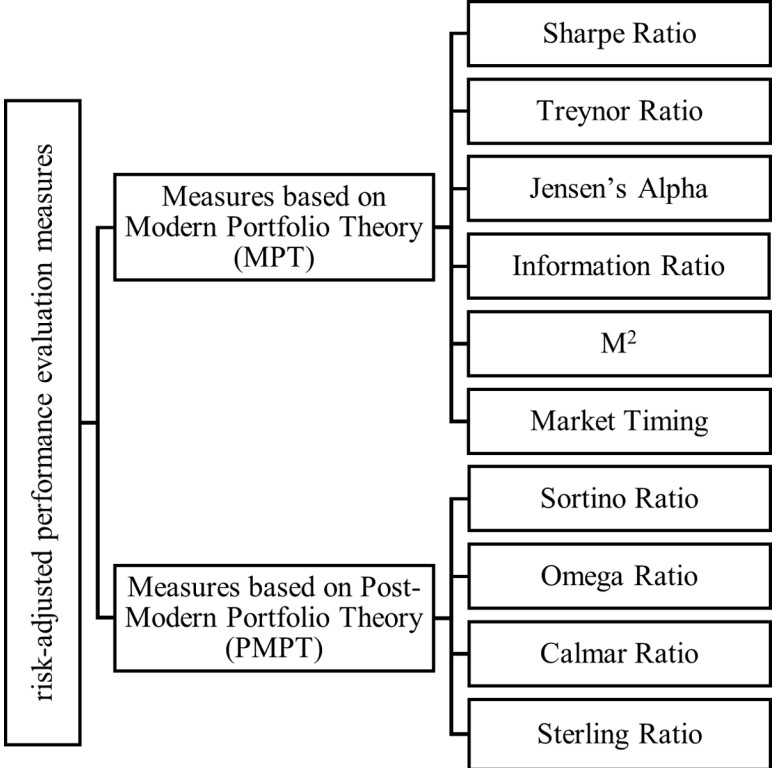

**Fig 2. Types of risk-adjusted performance evaluation measures.**

pertinent criteria and evaluation metrics. This study offers a comprehensive analysis, encapsulating the intricacies of each alternative. The detailed exposition of these criteria is presented in Table 1:

## 5. Computational results

In this section, we initially present the descriptive statistics pertaining to the monthly returns of each fund under scrutiny, as detailed in Table 2. The dataset, comprising daily observations from 14th January 2023–21st June 2024, facilitated the computation of monthly returns. The funds in question are designated as STARS, NAMAD, SANAM, and TMSK. TEDPIX (Tehran Exchange Dividend and Price Index), serving as a proxy for stock market performance, is employed as a benchmark index. Subsequently, we undertake a systematic and rigorous analysis of the outputs generated by the proposed methodology. Thereafter, the robustness and validity of the case study results are corroborated using a myriad of validation techniques. Initially, we juxtapose the ranking outcomes derived from the case study with those obtained via conventional Multi-Criteria Decision-Making methodologies prevalent in the extant literature. Subsequently, the sensitivity analysis of the ranking results is conducted by introducing perturbations to the input weights, thereby assessing the stability and resilience of the rankings against variations in the input parameters.

### 5.1. Step-by-step analysis of the outputs obtained by the entropy based WASPAS approach

The subsequent sections present a detailed exposition of the computational results obtained through the application of the WASPAS method. Each procedural step is meticulously explained, offering a comprehensive and scientifically rigorous analysis. This structured approach ensures clarity and facilitates a thorough understanding of the methodology, underscoring its utility in multi-criteria decision-making.

**Table 1. Introduction of performance evaluation criteria.**

| Criterion | Describe | | | |
|---|---|---|---|---|
| **Annualized Return** | An annualized total return is the geometric average amount of money an investment earns each year over a given period. | | | |
| | $R_a = (1 + R_m)^{\frac{12}{T}} - 1$ | | | (11) |
| | $R_a = Annualized\ Return$ | $R_m = Monthly\ Return$ | $T = number\ of\ months$ | |
| **Annualized Standard Deviation** | Annualized volatility extrapolates data over a year, enabling investors to compare the risk of investments with different return periods and time horizons. It is calculated by multiplying the standard deviation of returns by the square root of the number of periods in a year. | | | |
| | $\sigma_a = \sigma_m * \sqrt{12}$ | | | (12) |
| | $\sigma_a = Annualized\ Volatility$ | $\sigma_m = Monthly\ Volatility$ | | |
| **Jensen's Alpha** | Jensen's alpha is the average return on the portfolio over and above that predicted by the CAPM, given the portfolio's beta and the average market return [27]. $\alpha_j = R_p - R_f - \beta\ (R_m - R_f)$ | | | |
| | $R_p = Portfolio\ rate\ of\ return$ | $\beta = portfolio's\ beta$ | | (13) |
| | $R_f = Risk-free\ rate\ of\ return$ | $R_m = Market\ rate\ of\ return$ | | |
| **Beta** | The measure of systematic risk with respect to a benchmark. Systematic risk is the tendency of the value of the fund and the value of the benchmark to move together. | | | |
| | $\beta = \frac{COV(R_i, R_m)}{VAR(R_m)}$ | | | (14) |
| | $R_i = FOF\ rate\ of\ return$ | $R_m = Market\ rate\ of\ return$ | | |
| **Sharpe Ratio** | The Sharpe ratio divides the average portfolio excess return over the sample period by the standard deviation of returns over that period. It measures the reward to (total) volatility trade-off [28]. | | | |
| | $\textbf{Sharpe Ratio} = \frac{R_p - R_f}{\sigma_p}$ | | | (15) |
| | $R_p = Portfolio\ rate\ of\ return$ | $R_f = Risk-free\ rate\ of\ return$ | $\sigma_P = Portfolio\ Volatility$ | |
| **Sortino Ratio** | The Sortino Ratio, similar to the Sharpe Ratio, differs by focusing on downside risk instead of total volatility. It uses downside deviation as its risk measure, addressing the issue of treating upside volatility, which benefits investors, as part of total risk. A higher Sortino Ratio, like the Sharpe Ratio, indicates better performance[29]. | | | |
| | $\textbf{Sortino Ratio} = \frac{R_p - R_f}{\sigma_d}$ | | | (16) |
| | $R_p = Portfolio\ rate\ of\ return$ | $R_f = Risk-free\ rate\ of\ return$ | $\sigma_d = Portfolio\ Downside\ Volatility$ | |
| **Treynor Ratio** | Like the Sharpe ratio, Treynor's measure gives excess return per unit of risk, but it uses systematic risk instead of total risk [30]. | | | |
| | $\textbf{TR} = \frac{R_p - R_f}{\beta_p}$ | | | (17) |
| | $R_p = Portfolio\ rate\ of\ return$ | $R_f = Risk-free\ rate\ of\ return$ | $\beta_P = Portfolio\ Beta$ | |
| **Information Ratio** | The information ratio divides the alpha of the portfolio by nonsystematic risk, called "tracking error" in the industry. It measures abnormal return per unit of risk that in principle could be diversified away by holding a market index portfolio [31]. | | | |
| | $\textbf{IR} = \frac{\alpha_p}{\sigma_{e_p}}$ | | | (18) |
| | $\alpha_j = Jensen's\ Alpha$ | $\sigma_{e_p} = Tracking\ Error$ | | |
| **M2 (Modigliani ratio)** | Graham and Harvey, later popularized by Leah Modigliani and Franco Modigliani, proposed an equivalent representation of Sharpe's ratio called the M2 measure. Like the Sharpe ratio, M2 focuses on total volatility as a measure of risk but adjusts it to provide an easy-to-interpret differential return relative to the benchmark index. M2 is computed by mixing an active portfolio with T-bills to match the volatility of a passive market index [32]. | | | |
| | $M^2 = \frac{\sigma_m}{\sigma_p}(R_p - R_f) - (R_m - R_f)$ | | | (19) |
| | $R_p = Portfolio\ rate\ of\ return$ | $R_f = Risk-free\ rate\ of\ return$ | $R_m = Market\ rate\ of\ return$ | |
| | $\sigma_m = Market\ Volatility$ | $\sigma_P = Portfolio\ Volatility$ | | |

*(Continued)*

**Table 1.** (Continued)

| Criterion | Describe | | |
|---|---|---|---|
| **Morningstar (MRAR)** | The Morningstar Risk-Adjusted Return (MRAR) is calculated by subtracting Morningstar Risk, which emphasizes downside variation, from Morningstar Return, based on historical excess returns. MRAR is used to determine the Morningstar Rating for funds, comparing each investment's MRAR against others in its category to assign a star rating [33]. | | |
| | $$\text{MRAR } (\gamma) = \left[ \frac{1}{T} \sum_{t=1}^{T} \left( \frac{1+r_t}{1+r_{ft}} \right)^{-\gamma} \right]^{\frac{12}{\gamma}} - 1$$ | | (20) |
| | $T$ = number of months in a period | $r_t$ = the return of the fund in month t | |
| | $\gamma$ = investor's level of risk aversion | $r_{ft}$ = the return on risk free asset in month t | |
| **Omega Ratio** | Can be used as an alternative to the Sharpe Ratio in measuring risk-adjusted return. Omega is defined by Shadwick and Keating [2002], and unlike an investment's Sharpe, Omega doesn't assume a normal return distribution. It focuses on the likelihood of not meeting a target return [34]. | | |
| | $$\Omega = \frac{\frac{1}{n} * \sum_{i=1}^{i} \max(r_i - r_{T,0})}{\frac{1}{n} * \sum_{i=1}^{n} \max(r_T - r_{i,0})}$$ | | (21) |
| | $r_i$ = FOF rate of return | $r_T$ = minimum expected return | |
| **Market Timing Treynor & Mazuy** | Market timing is a source of portfolio risk variation, involving shifts between a market-index portfolio and a safe asset based on expected performance [35]. Most managers do not fully shift between T-bills and the market. Regression analysis estimates parameters a, b, and c, and if c is positive, it indicates timing ability, as the characteristic line steepens with larger $R_m - R_f$ | | |
| | $R_p - R_f = a + b\,(R_m - R_f) + c\,(R_m - R_f)^2 + e_p$ | | (22) |
| | $R_p$ = Portfolio rate of return | $R_m$ = Market rate of return | |
| | $R_f$ = Risk − free rate of return | $e_P$ = Portfolio Error | |
| **Calmar Ratio** | The Calmar Ratio, a variation of the Sterling Ratio, measures an investment's return relative to drawdown, commonly used with hedge funds. A higher Calmar Ratio indicates better investment performance. Maximum Drawdown is the portfolio's maximum loss from a peak to a trough before a new peak is reached [36]. | | |
| | $\text{CR} = \frac{R_p - R_f}{\text{Maximum Drawdown}}$<br><br>$\text{MDD} = \frac{\text{Trough value} - \text{Peak value}}{\text{Peak Value}}$ | | (23) |
| | $R_p$ = Portfolio rate of return | $R_f$ = Risk − free rate of return | |
| **Sterling Ratio** | Used mainly in the context of hedge funds, this risk-reward measure determines which hedge funds have the highest returns while enduring the least amount of volatility | | |
| | $\text{Sterling Ratio} = \frac{\text{Compounded Annual Return}}{\text{Average Maximum Drawdown} - 10\%}$ | | (24) |

**Step 1:** *Formulation of the Decision Matrix*

In this phase, we construct a decision matrix where each element encapsulates the performance metric of an alternative with respect to a specific criterion. This matrix serves as the foundational framework, systematically organizing all performance metrics and facilitating an exhaustive evaluation of all alternatives across multiple criteria. The decision matrix for this specific case study, which displays the performance of the alternatives concerning the considered criteria, is provided in Table 3.

**Step 2:** *Normalization of the Decision Matrix*

To ensure rigorous uniformity and comparability, we implement normalization on the constituents of the decision matrix. This process obliterates dimensional attributes, thereby standardizing the criteria measurements and eliminating discrepancies in scale. The normalization technique is contingent upon the nature of the criterion, whether it is beneficial (where elevated values are preferable) or non-beneficial (where diminished values are preferable). Table 4 elucidates the

**Table 2. Descriptive statistics of monthly returns for selected funds.**

| Descriptive statistics | STARS | NAMAD | SANAM | TMSK | TEDPIX |
|---|---|---|---|---|---|
| Mean | 0.0122 | 0.0173 | 0.0160 | 0.0285 | 0.0155 |
| Standard Error | 0.0044 | 0.0251 | 0.0157 | 0.0182 | 0.0240 |
| Median | 0.0064 | -0.0045 | 0.0064 | 0.0122 | -0.0196 |
| Standard Deviation | 0.0180 | 0.1037 | 0.0649 | 0.0749 | 0.0988 |
| Sample Variance | 0.0003 | 0.0107 | 0.0042 | 0.0056 | 0.0098 |
| Kurtosis | -0.6524 | 3.8459 | 2.3384 | 1.4558 | 2.0897 |
| Skewness | 0.4059 | 1.7451 | 1.2352 | 1.0636 | 1.5129 |
| Range | 0.0620 | 0.4285 | 0.2687 | 0.2905 | 0.3708 |
| Minimum | -0.0143 | -0.1055 | -0.0770 | -0.0705 | -0.0922 |
| Maximum | 0.0477 | 0.3230 | 0.1917 | 0.2201 | 0.2785 |
| Quantile 1 | 0.0005 | -0.0613 | -0.029 | -0.0213 | 0.0112 |
| Quantile 2 | 0.0064 | -0.0045 | 0.0064 | 0.0122 | -0.0196 |
| Quantile 3 | 0.0272 | 0.0742 | 0.0474 | 0.0683 | 0.0160 |
| | | | | | |

**Table 3. Constructed decision matrix for performance metrics.**

| Criterion | Performance | STARS | NAMAD | SANAM | TMSK | TEDPIX |
|---|---|---|---|---|---|---|
| Annualized Return | Higher | 0.155 | 0.164 | 0.183 | 0.361 | 0.144 |
| Jensen Alpha | Higher | (0.015) | 0.021 | 0.027 | 0.206 | – |
| Sharpe | Higher | (0.311) | (0.029) | 0.038 | 0.718 | (0.089) |
| Sortino | Higher | (0.925) | (0.093) | 0.091 | 1.999 | (0.309) |
| Treynor | Higher | (0.124) | (0.010) | 0.014 | 0.281 | (0.031) |
| Information | Higher | (0.457) | 0.212 | 0.309 | 1.643 | – |
| M2 (Modigliani) | Higher | (0.076) | 0.021 | 0.044 | 0.276 | – |
| Omega | Higher | 6.939 | 1.648 | 2.109 | 3.025 | 1.578 |
| Sterling | Higher | 1.512 | 1.044 | 1.299 | 2.572 | 0.824 |
| Calmar | Higher | (0.949) | (0.064) | 0.086 | 1.545 | (0.184) |
| MRAR | Higher | (0.020) | (0.100) | -0.035 | 0.095 | (0.111) |
| Treynor-Mazuy | Higher | (0.298) | 1.035 | 0.562 | 0.905 | – |

normalization of the decision matrix concerning performance metrics, offering an exhaustive overview of the standardization methodology employed.

**Step 3:** *Entropy-Based Weighting*

Subsequently, we compute the entropy for each criterion to determine their respective weights. This process involves formulating a probability matrix based on the normalized decision matrix, followed by the calculation of entropy for each criterion to ascertain the degree of diversification. The weights are derived from these diversification metrics, indicating the relative significance of each criterion within the decision-making framework. The values are delineated in Table 5.

**Step 4:** *Computation of the Aggregate Weight Using the Weighted Sum Model (WSM).*

We then calculate an aggregate weight for each alternative using the Weighted Sum Model, as presented in Table 6. This entails summing the normalized scores of each criterion, weighted by their respective importance, thus providing an initial quantitative measure of each alternative's overall performance.

**Step 5:** *Evaluation of the Relative Importance Using the Weighted Product Model (WPM).*

**Table 4. Normalization of the decision matrix for performance metrics.**

| Criterion | STARS | NAMAD | SANAM | TMSK |
|---|---|---|---|---|
| *Annualized Return* | 0.000 | 0.044 | 0.136 | 1.000 |
| *Jensen Alpha* | 0.000 | 0.163 | 0.190 | 1.000 |
| *Sharpe* | 0.000 | 0.274 | 0.339 | 1.000 |
| *Sortino* | 0.000 | 0.285 | 0.347 | 1.000 |
| *Treynor* | 0.000 | 0.281 | 0.341 | 1.000 |
| *Information* | 0.000 | 0.319 | 0.365 | 1.000 |
| *M2 (Modigliani)* | 0.000 | 0.276 | 0.341 | 1.000 |
| *Omega* | 1.000 | 0.000 | 0.087 | 0.260 |
| *Sterling* | 0.306 | 0.000 | 0.167 | 1.000 |
| *Calmar* | 0.000 | 0.355 | 0.415 | 1.000 |
| *MRAR* | 0.410 | 0.000 | 0.333 | 1.000 |
| *Treynor-Mazuy* | 0.000 | 1.000 | 0.645 | 0.902 |

**Table 5. Entropy values and objective weights for performance metrics.**

| Criterion | $H_j$ | $d_j$ | $w_j$ |
|---|---|---|---|
| *Annualized Return* | 0.369 | 0.631 | 0.143 |
| *Jensen Alpha* | 0.544 | 0.456 | 0.104 |
| *Sharpe* | 0.668 | 0.332 | 0.076 |
| *Sortino* | 0.674 | 0.326 | 0.074 |
| *Treynor* | 0.671 | 0.329 | 0.075 |
| *Information* | 0.689 | 0.311 | 0.071 |
| *M2 (Modigliani)* | 0.669 | 0.331 | 0.075 |
| *Omega* | 0.516 | 0.484 | 0.11 |
| *Sterling* | 0.603 | 0.397 | 0.09 |
| *Calmar* | 0.710 | 0.29 | 0.066 |
| *MRAR* | 0.704 | 0.296 | 0.067 |
| *Treynor-Mazuy* | 0.781 | 0.219 | 0.05 |

In this phase, we evaluate the relative importance of each alternative using the Weighted Product Model, as detailed in Table 7. This method involves the multiplicative aggregation of normalized scores, each raised to the power of their respective weights. This highlights the proportional contribution of each criterion, offering a nuanced measure of each alternative's relative importance.

**Step 6:** *Integration of WSM and WPM Scores*

We amalgamate the scores derived from both the WSM and the WPM to formulate a comprehensive metric for each alternative. This synthesis ensures an equitable representation from both models, culminating in a balanced and holistic evaluation of the alternatives. The resultant values are tabulated in Table 8.

**Step 7:** *Generalized Aggregated Criterion*

To facilitate a dynamic equilibrium between the Weighted Sum Model and the Weighted Product Model, we have formulated a generalized aggregated criterion. This entails the implementation of a weighting parameter that modulates the balance between the two models. This malleability enables decision-makers to calibrate the equilibrium in accordance with distinct decision-making frameworks or preferences. In pursuit of this objective, we adjust the Lambda (λ) parameter

**Table 6. Aggregate scores computed using the weighted sum model (WSM).**

| Score | STARS | NAMAD | SANAM | TMSK |
|-------|-------|-------|-------|------|
| *WSM* | 0.165041 | 0.202300 | 0.273860 | 0.913898 |

**Table 7. Aggregate scores computed using the weighted product model (WPM).**

| Score | STARS | NAMAD | SANAM | TMSK |
|-------|-------|-------|-------|------|
| *WPM* | 0.000000 | 0.000000 | 0.238397 | 0.858171 |

**Table 8. Combined scores derived from the integration of WSM and WPM scores.**

| Score | STARS | NAMAD | SANAM | TMSK |
|-------|-------|-------|-------|------|
| *WSM and WPM* | 0.082521 | 0.101150 | 0.256128 | 0.886035 |

**Table 9. Impact of λ on performance ranking.**

| Weighting parameter | STARS | NAMAD | SANAM | TMSK |
|---------------------|-------|-------|-------|------|
| $\lambda = 0$ | 0.000000 | 0.000000 | 0.238397 | 0.858171 |
| $\lambda = 0.1$ | 0.016504 | 0.020230 | 0.241943 | 0.863744 |
| $\lambda = 0.2$ | 0.033008 | 0.040460 | 0.245489 | 0.869316 |
| $\lambda = 0.3$ | 0.049512 | 0.060690 | 0.249036 | 0.874889 |
| $\lambda = 0.4$ | 0.066016 | 0.080920 | 0.252582 | 0.880462 |
| $\lambda = 0.5$ | 0.082521 | 0.101150 | 0.256128 | 0.886035 |
| $\lambda = 0.6$ | 0.099025 | 0.121380 | 0.259674 | 0.891607 |
| $\lambda = 0.7$ | 0.115529 | 0.141610 | 0.263221 | 0.897180 |
| $\lambda = 0.8$ | 0.132033 | 0.161840 | 0.266767 | 0.902753 |
| $\lambda = 0.9$ | 0.148537 | 0.182070 | 0.270313 | 0.908325 |
| $\lambda = 1$ | 0.165041 | 0.202300 | 0.273860 | 0.913898 |

from 0 to 1, allowing for the evaluation of the Waspas model outcomes in the context of fund assessments. The resultant data are delineated in Table 9.

The Impact of λ on the Performance Evaluation Matrix of the WASPAS Model provides an exhaustive comparative assessment of four alternatives STARS, NAMAD, SANAM, and TMSK across a spectrum of weighting parameters spanning from 0 to 1. The λ parameter, a pivotal component of the WASPAS methodology, adjusts the equilibrium between the Weighted Sum Model and the Weighted Product Model, thereby affecting the resultant performance rankings of the examined alternatives.

Empirical data indicates that the performance rankings of these alternatives demonstrate a notable degree of stability and robustness in response to variations in λ. This robustness signifies that the model's sensitivity to changes in λ is minimal, thereby ensuring consistent decision-making reliability. Such stability is essential for decision-makers as it highlights the model's resilience to parameter fluctuations, thereby reinforcing confidence in the accuracy of the rankings produced through the WASPAS method. Consequently, the stability of the ranking matrix reduces the risk of erroneous decision-making due to parameter sensitivity, enhancing the model's applicability across diverse decision-making contexts.

## 5.2. Validation of results

Multiple Attribute Decision Making methodologies endeavor to produce dependable and consistent outcomes. However, the resulting rankings can be susceptible to various determinants. These determinants encompass modifications to criterion weights, the inclusion or exclusion of alternatives, subjective evaluations, and the precise selection of criteria. Consequently, this section undertakes the validation of the case study results through diverse methodologies. In Section 5.2.1., the ranking outcomes of the case study are juxtaposed with those generated by prevalent MADM techniques documented in extant literature, and the correlations between them are scrutinized. In Section 5.2.2., the sensitivity of the derived ranking results to fluctuations in input weights is rigorously examined.

### 5.2.1. Ranking results of other methods.

In this section, the researchers undertook a rigorous comparative analysis between the outcomes derived from the case study, wherein the weights were ascribed via entropy, and the outcomes procured through various sophisticated ranking methodologies. The paramount objective was to meticulously assess the concordance or discordance of the ultimate rankings in relation to those produced by alternative methods. The transformed decision matrix underwent scrutiny through an array of ranking techniques, encompassing the Technique for Order of Preference by Similarity to Ideal Solution (TOPSIS) [37], the Weighted Aggregated Sum Product Assessment (WASPAS) [19], the Additive Ratio Assessment (ARAS) [38], the Measurement of Alternatives and Ranking according to Compromise Solution (MARCOS) [39], the Evaluation based on Distance from Average Solution (EDAS) [40], the Multi-Objective Optimization on the basis of Ratio Analysis (MOORA) [41], the Complex Proportional Assessment (COPRAS) [42], the Multi-Attributive Border Approximation area Comparison (MABAC) [43], and the Measuring Attractiveness by a Categorical Based Evaluation Technique (MACBETH) [44].

The Multi-Criteria Decision Making web-service computation engine (https://www.mcdm.app/) was utilized to generate the results of these methodologies. The findings, delineated in Table 10, elucidate the variances in rankings. Moreover, by leveraging the Augmented WASPAS methodology, this study proffers a robust framework for appraising the efficiency of Fund of Funds, incorporating risk-adjusted performance metrics. The amalgamation of diverse ranking methods ensures a thorough evaluation, addressing the intrinsic complexities of the decision-making process in fund management.

The comparative analysis of disparate multi-criteria decision-making (MCDM) methodologies elucidates significant discrepancies in the rankings of four alternatives based on specific evaluative criteria. For instance, TOPSIS assigns alternative A2 a second-place ranking, whereas WASPAS assigns it a fourth-place ranking. Similarly, COPRAS and MABAC rank alternative A4 as third and fourth, respectively. These divergences in utility valuations underscore the substantial influence that the selection of a particular decision-making method can exert on the ultimate determination. The observed discrepancies in rankings stem from the fundamental differences in each methodology's mathematical structure and underlying evaluation principles. Each approach, including WASPAS, ARAS, MARCOS, EDAS, MOORA, COPRAS, MABAC,

**Table 10. Ranking results of other MADM approaches.**

| Alternatives | WASPAS | Rank | ARAS | Rank | MARCOS | Rank | EDAS | Rank | MOORA | Rank |
|---|---|---|---|---|---|---|---|---|---|---|
| $A_1$ | 0.39 | 2 | -0.08 | 3 | 1.13 | 2 | 0.52 | 1 | 0.07 | 2 |
| $A_2$ | 0.30 | 4 | 1.00 | 1 | 3.26 | 1 | 0.08 | 3 | 0.04 | 3 |
| $A_3$ | 0.34 | 3 | -0.66 | 4 | -2.55 | 4 | 0.07 | 4 | 0.04 | 4 |
| $A_4$ | 0.84 | 1 | 0.02 | 2 | -0.11 | 3 | 0.50 | 2 | 0.11 | 1 |
| Alternatives | COPRAS | Rank | MABAC | Rank | MACBETH | Rank | TOPSIS | Rank | Augmented Waspas | Rank |
| $A_1$ | -0.67 | 4 | 0.28 | 1 | 16.49 | 4 | 0.77 | 1 | 0.08 | 4 |
| $A_2$ | 0.74 | 2 | 0.24 | 2 | 20.27 | 3 | 0.74 | 2 | 0.10 | 3 |
| $A_3$ | 0.77 | 1 | 0.17 | 3 | 27.40 | 2 | 0.68 | 3 | 0.26 | 2 |
| $A_4$ | 0.17 | 3 | -0.47 | 4 | 91.38 | 1 | 0.16 | 4 | 0.89 | 1 |

MACBETH, and TOPSIS, employs distinct weighting schemes and aggregation mechanisms that place varying degrees of emphasis on particular aspects of the decision criteria. Consequently, these unique methodological attributes lead to different prioritization and ranking outcomes across alternatives when comparing them with the augmented WASPAS approach. This variability underscores the need to carefully consider the choice of method, as it can critically shape the decision-making process and the conclusions drawn from it. In this paper, the WASPAS methodology was selected due to its robust multi-criteria decision-making framework, which effectively balances both additive and multiplicative aggregation approaches. This dual approach allows WASPAS to capture a comprehensive picture of alternative performances, making it particularly well-suited for evaluating complex investment structures such as Fund of Funds. However, to address certain limitations inherent in the traditional WASPAS model such as its sensitivity to weighting schemes and its limited ability to identify the most influential performance criteria an augmented version of WASPAS was developed. This enhancement incorporates entropy-based weighting and refined evaluative techniques, thereby improving its robustness, precision, and ability to provide a more nuanced assessment of Fund of Funds performance.

**5.2.2. Weight sensitivity analysis.** The primary goal of conducting a sensitivity analysis within a specific MADM algorithm is to assess the influence of variations in given conditions on the resulting rankings. The purpose is to evaluate the stability of the outcomes generated by the MADM method. One of the key factors that can introduce fluctuations in the given conditions and impact the results of MADM methods is changes in criteria weights. This section focuses on investigating the sensitivity of the results by deliberately modifying the input weight matrix In this particular case study, where subjective weights determined by decision-makers are of particular importance, a percentage of error is introduced into the values of these subjective weights to analyze its impact on the overall performance of the available options. For example, let's assume that we want to assess the sensitivity of the outcomes to changes in $w_j^s$ (the subjective weights). By assigning a percentage of change and error in $w_j^s$ as $er$, we can obtain the modified weight denoted as $w_j^{ms}$ using the following relationship:

$$w_j^{ms} = (1 - \frac{er}{100})w_j^s \tag{25}$$

The calculation of other modified attribute weights resulting from changes in $w_j^s$ is determined as follows:

$$w_i^s = \left( \frac{1 - (1 - \frac{er}{100})w_j^s}{1 - w_j^s} \right) w_i^s \; \forall i = 1, 2, \ldots, m. \; i \neq j \tag{26}$$

To analyze the sensitivity, we examine 4 different values for $er$, which represents the degree of change or error in the weights.

$$er \in \{-50\%, -25\%, \; 25\%, \; 50\%\}$$

The overall performance of all alternatives is computed for each percentage error value and modified weights. The scatter plot of these data points is depicted in Fig 3.

Fig 3 elucidates the ramifications of minute weight perturbations on the aggregate performance and hierarchical ordering of distinct alternatives within the refined WASPAS methodology. Despite observable oscillations in the performance metrics of each alternative, the terminal ranking remains invariant across all sensitivity analyses pertaining to weight adjustments. This invariability underscores the robustness of the methodology concerning weight sensitivity, suggesting that criterion weights exert a negligible influence on the determination of the optimal alternative. This invariability fosters enhanced confidence in both the decision-making process and the selected alternative. The capacity to sustain stability amidst minor weight fluctuations is a commendable attribute of the proposed methodology, indicating that the hierarchical

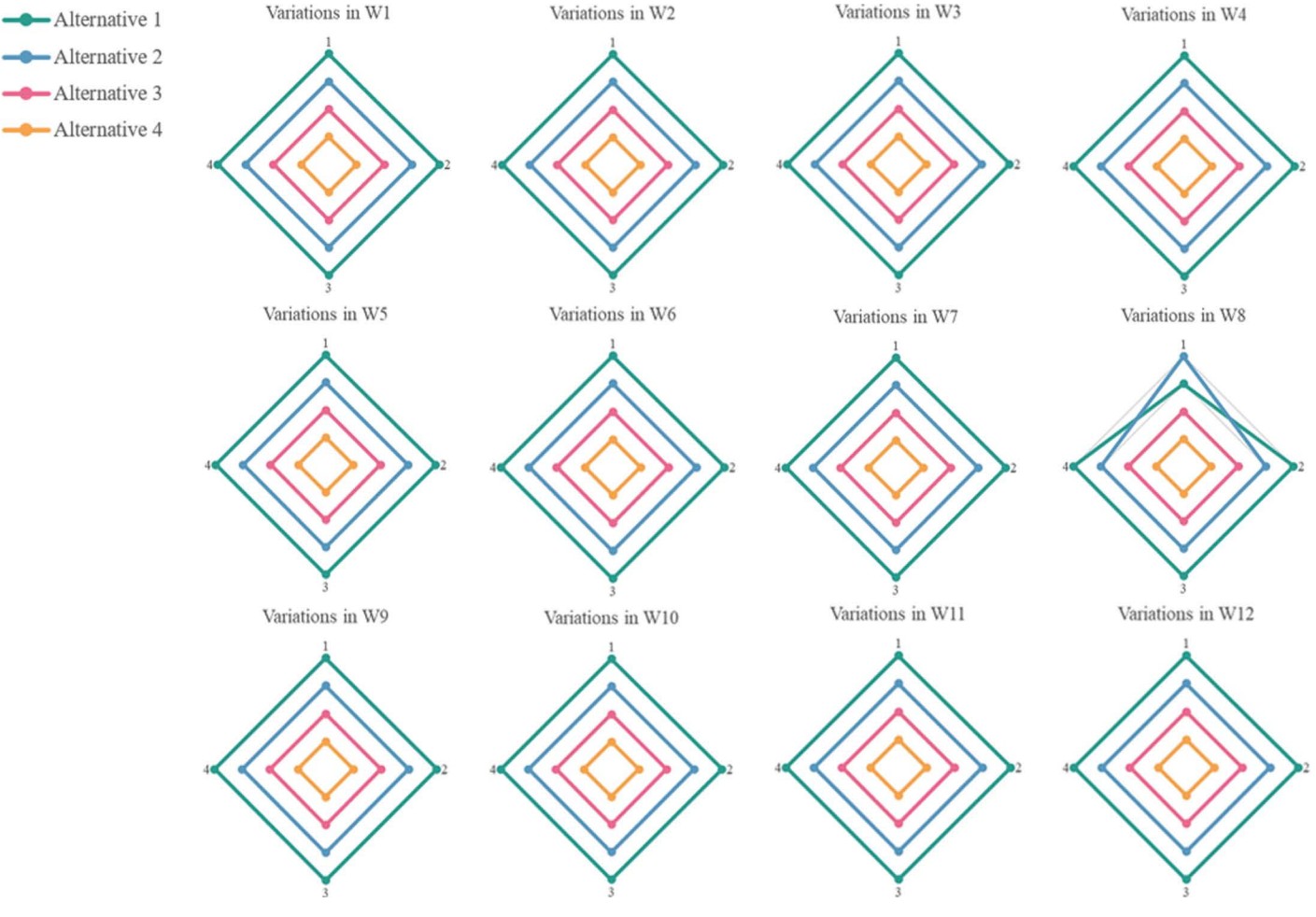

**Fig 3. Weight sensitivity analysis.**

rankings of alternatives exhibit relative constancy and are not significantly swayed by trivial variations in criterion weights. This further bolsters confidence in the decision-making process and the ultimate choice of the alternative.

### 5.3. Discussion and managerial implications

The study introduces an innovative augmentation to the WASPAS methodology, designed to refine the evaluative mechanisms for Fund of Funds performance by leveraging risk-adjusted metrics. This enhancement provides a comprehensive and sophisticated decision-making framework. The incorporation of entropy for weight calculation significantly enhances precision and reliability, effectively addressing the inherent complexities involved in assessing multi-layered investment structures. This methodological advancement holds profound implications for managerial practices in portfolio management. By employing quantitative attributes, the modified WASPAS approach transcends the limitations of traditional evaluation models, offering a nuanced and robust assessment tool.

The empirical application in the case study vividly demonstrates the efficacy of this approach, revealing significant variations in fund performance that conventional methods might overlook. The entropy-based weighting mechanism significantly augments the objectivity of the evaluation process, ensuring that the derived weights accurately reflect the relative

importance of each criterion. This objective weighting is crucial in mitigating biases and ensuring that the performance metrics are evaluated on a uniform scale.

Furthermore, the dual aggregation process, which amalgamates the WSM and the WPM, synthesizes the advantages of both additive and multiplicative approaches, providing a balanced and holistic evaluation of the alternatives. This approach enhances the comprehensiveness of the evaluation process, making it more robust and reliable.

The sensitivity analysis underscores the robustness of the modified WASPAS methodology, demonstrating its resilience to variations in criteria weights. This stability is vital for managerial decision-making, instilling confidence that the rankings of fund performance remain consistent under different weighting scenarios. Such reliability is indispensable for strategic planning and resource allocation in investment management. From a practical perspective, the transparency and methodological rigor of the modified WASPAS approach facilitate its adoption in real-world applications. The step-by-step explication ensures that practitioners can easily comprehend and implement the methodology, making it an invaluable tool for evaluating complex investment structures.

This methodological clarity, coupled with its robust analytical capabilities, positions the modified WASPAS method as a superior evaluative framework in the financial domain. In conclusion, the proposed modifications to the WASPAS methodology represent a significant advancement in multi-attribute decision-making, particularly in the context of fund-of-funds evaluation. The enhanced precision, robustness, and practical applicability of this approach provide a formidable tool for investment managers, enabling more informed and strategic decision-making. The study's findings underscore the utility of this refined methodology in navigating the complexities of modern portfolio management, ensuring optimal performance evaluation and strategic alignment with financial objectives.

Overall, the study offers a significant contribution to the field of financial evaluation and portfolio management. The methodological innovations introduced have the potential to transform how investment performance is assessed, making the process more accurate, reliable, and applicable to the nuanced realities of modern financial markets. This advancement not only aids in better decision-making but also contributes to the broader understanding and development of sophisticated evaluation techniques in the finance industry.

## 6. Conclusion

The intensifying demand for refined investment methodologies and the volatile nature of financial markets impose substantial obstacles in our quest for enduring and lucrative portfolio administration. FOF, as a strategic investment mechanism, proffers an efficacious resolution for diversification and risk mitigation due to its proficiency in capital allocation across multiple subordinate funds. These attributes render FOFs as a sustainable investment strategy indispensable for realizing long-term financial objectives. Nonetheless, ascertaining the most efficacious FOF necessitates a thorough analysis.

To mitigate this challenge, deploying Multi-Attribute Decision Making methodologies offers a viable resolution. Consequently, this study employs a MADM technique to appraise the performance of various FOF strategies. Four distinct FOF alternatives and twelve performance criteria are considered in the evaluation. The study meticulously assesses the performance of the selected FOF, encompassing economic, risk, return, diversification, and management quality dimensions.

The principal findings of this study can be encapsulated as follows: After appraising all criteria for evaluating and juxtaposing the performance of various FOF strategies, TMSK emerges as the most efficacious option.

It is succeeded by SANAM, which also ranks prominently in terms of performance. In contrast, STARS is adjudged the least efficacious choice. These findings are invaluable for investors, fund managers, financial advisors, policymakers, and researchers. They can leverage this information to formulate and implement an investment blueprint that fosters more sustainable and profitable portfolio management. Moreover, the comparative analysis of the proposed method against other MADM techniques and the conducted sensitivity analysis corroborate the efficiency and robustness of the proposed framework.

The modified WASPAS framework bears practical implications for FOF investments, furnishing a systematic approach for resource allocation and adherence to industry standards. Furthermore, it advocates interdisciplinary collaboration, potentially establishing a substantive knowledge base for identifying an apt approach to FOF strategies.

Future research can delve deeper into the following domains: Firstly, while this study concentrates on comparatively assessing the performance of various FOF strategies to augment investment outcomes, future research can broaden this approach to explore the commercialization and market proliferation of these strategies. This includes examining facets such as infrastructure and market expansion, viable product development, and the sustainability of FOF investments. Secondly, it is recommended to incorporate additional performance criteria into the evaluation process. These may encompass factors such as system longevity, response time, liquidity, and suitability for diverse market conditions. Additionally, more sophisticated financial instruments and innovative investment strategies should be contemplated in the performance assessment. By exploring these research areas, a more comprehensive understanding of FOF strategies can be attained, leading to the development of more efficacious investment techniques and technologies in this field.

## Supporting information

**S1 Data. Raw net asset value (NAV) data used for the risk-adjusted performance calculations.**
(CSV)

## Author contributions

**Conceptualization:** Mostafa Shabani.

**Data curation:** Mostafa Shabani, Ali Khodarahmi.

**Formal analysis:** Mostafa Shabani, Ali Khodarahmi.

**Funding acquisition:** Mostafa Shabani.

**Methodology:** Mostafa Shabani.

**Resources:** Mostafa Shabani.

**Supervision:** Rouzbeh Ghousi, Emran Mohammadi.

**Validation:** Mostafa Shabani, Ali Khodarahmi.

**Visualization:** Mostafa Shabani, Ali Khodarahmi.

**Writing – original draft:** Mostafa Shabani, Hossein Ghanbari.

**Writing – review & editing:** Mostafa Shabani, Hossein Ghanbari.

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
