## [Decision Letter · Decision Letter 0]

5 Nov 2024

Dear Dr. Ghousi,

Thank you for submitting your manuscript to PLOS ONE. After careful consideration, we feel that it has merit but does not fully meet PLOS ONE’s publication criteria as it currently stands. Therefore, we invite you to submit a revised version of the manuscript that addresses the points raised during the review process.

**ACADEMIC EDITOR:**

We look forward to receiving your revised manuscript.

Kind regards,

Gabrijela Popovic

Academic Editor

PLOS ONE

Journal Requirements:

2. Please note that your Data Availability Statement is currently missing [the repository name and/or the DOI/accession number of each dataset OR a direct link to access each database]. If your manuscript is accepted for publication, you will be asked to provide these details on a very short timeline. We therefore suggest that you provide this information now, though we will not hold up the peer review process if you are unable.

Reviewers' comments:

Reviewer's Responses to Questions

**Comments to the Author**

1. Is the manuscript technically sound, and do the data support the conclusions?

Reviewer #1: Yes

Reviewer #2: Yes

2. Has the statistical analysis been performed appropriately and rigorously?

Reviewer #1: Yes

Reviewer #2: N/A

3. Have the authors made all data underlying the findings in their manuscript fully available?

Reviewer #1: Yes

Reviewer #2: Yes

4. Is the manuscript presented in an intelligible fashion and written in standard English?

Reviewer #1: Yes

Reviewer #2: Yes

Reviewer #1: Dear authors,

The submitted study delivers valuable insights through original research findings that have not previously been published.

The title clearly describes the topic, and according to the journal's guidelines, it has less than 250 characters.

The article is well written in standard American English, though a few minor issues could be addressed for enhanced clarity. The existing literature could be better cited because the article lacks the most recent respectable articles. For example, there are no papers from 2024, only two from 2023, only one from 2022, no references from 2021 and only two references from 2020. Altogether, the article has only five references from the last five years. Authors should consult more recent literature to make their research up to date.

The abstract is written clearly without references and has 226 words, less than the maximum length. The abstract describes the main objective of the research and explains the methodology without too many details. According to the journal's instructions, the abbreviations should not be included if possible. The recommendation for authors is to modify the abstract and retain fewer abbreviations. It is clear that some of the abbreviations have to remain, but I firmly believe that authors can rewrite the abstract to include less than 12 abbreviations as they have in the abstract at the moment.

The introduction is written like a funnel, providing broad information regarding the research issue and then narrowing it. However, the introduction is almost four pages long, and the authors should consider dividing it. It also lacks referencing methods and models in the text currently in the introduction. The methodology is conducted to a respectable technical standard and is documented adequately. However, the authors should elaborate more on their choice of the WASPAS method rather than some of the more recent methods. The conclusions drawn are clearly articulated and well-supported by the data presented.

Importantly, no pertinent discussion was needed regarding the ethics of experimentation and research integrity, and I appreciate that the authors have made the data available for further validation.

Several segments could be improved regarding the formatting of the paper. For example, Table 8 should be written in capital first letter. There needs to be an introduction of Figure 2 in the integral text. Numbers in Figure 3 are written in different fonts. Numbers 2 and 4 should be further away from the chart to be observed more clearly. The manuscript should be formatted by the instructions to meet the journal's formatting requirements.

There are also specific issues in the reference list. Several references are missing certain segments. For example, Keating & Shadwick (2002), Treynor (1965), Treynor (1966), Young (1991) and several others need pagination; Fox (2016) needs to include the name of the book, etc. Authors are advised to check references and unify the list. Some references are written in all caps, some with capitalisations of titles and others regularly. All references have to be written according to the journal rules.

Overall, this study contributes significantly to the field, and I look forward to seeing the revised version.

Reviewer #2: While the article's concept and the proposed WASPAS approach are intriguing, there are some deficiencies present in the article:

1-) The studies employing the WASPAS approach are not elaborated upon comprehensively.

2-)The benefits of the Augmented WASPAS approach compared to the WASPAS method are not thoroughly delineated.

3-) Examination of Table 10 reveals significant discrepancies in the rankings. The discrepancies require elucidation.

**Do you want your identity to be public for this peer review?** For information about this choice, including consent withdrawal, please see our Privacy Policy

Reviewer #1: **Yes: ** Vuk Mirčetić

Reviewer #2: No

---

## [Author Response · Author response to Decision Letter 1]

16 Nov 2024

Dear Academic Editor Gabrijela Popovic.

We have submitted a thoroughly revised version of our manuscript, titled "An Appraisal of Fund of Funds Efficiency Based on Risk-Adjusted Performance Measures: Application of an Augmented WASPAS Methodology," for your consideration for publication in PLOS ONE.

We extend our profound gratitude to the reviewers for their incisive and invaluable feedback, which has played a pivotal role in enhancing the rigor and quality of our manuscript. Accompanying this letter are meticulously crafted responses to each of the reviewers' comments. Additionally, we have implemented comprehensive revisions aimed at elevating both clarity and precision. These modifications encompass refinements to the textual narrative and an enriched literature review, effectively addressing areas that previously exhibited ambiguity or lacked completeness.

We eagerly await your response at your earliest convenience.

Sincerely yours,

Rouzbeh Ghousi, Associate Professor

---

## [Editor Report · Decision Letter 1]

19 Nov 2024

An Appraisal of Fund of Funds Efficiency Based on Risk-Adjusted Performance Measures: Application of an Augmented WASPAS Methodology

PONE-D-24-42417R1

Dear Dr. Rouzbeh Ghousi, 

We’re pleased to inform you that your manuscript has been judged scientifically suitable for publication and will be formally accepted for publication once it meets all outstanding technical requirements.

Kind regards,

Gabrijela Popovic

Academic Editor

PLOS ONE

Additional Editor Comments (optional):

The authors successfully met the reviewer's requirements and addressed all aroused concerns, which improved the article's quality and made it suitable for publication.
---

## [Editor Report · Acceptance letter]

PONE-D-24-42417R1

PLOS ONE

Dear Dr. Ghousi,

I'm pleased to inform you that your manuscript has been deemed suitable for publication in PLOS ONE. Congratulations! Your manuscript is now being handed over to our production team.

Kind regards,

on behalf of

Dr. Gabrijela Popovic

Academic Editor

PLOS ONE